# Peripheral Inflammatory Markers TNF-α and CCL2 Revisited: Association with Parkinson’s Disease Severity

**DOI:** 10.3390/ijms24010264

**Published:** 2022-12-23

**Authors:** Georgia Xiromerisiou, Chrysoula Marogianni, Ioannis C. Lampropoulos, Efthimios Dardiotis, Matthaios Speletas, Panagiotis Ntavaroukas, Anastasia Androutsopoulou, Fani Kalala, Nikolaos Grigoriadis, Stamatia Papoutsopoulou

**Affiliations:** 1Department of Medicine, Faculty of Life Sciences, University of Thessaly, 41500 Larisa, Greece; 2Respiratory Medicine Department, Faculty of Medicine, University of Thessaly, 41500 Larissa, Greece; 3Department of Immunology & Histocompatibility, Faculty of Medicine, University of Thessaly, 41500 Larissa, Greece; 4Department of Biochemistry and Biotechnology, Faculty of Life Sciences, University of Thessaly, 41500 Larisa, Greece; 5Laboratory of Experimental Neurology and Neuroimmunology, Second Department of Neurology, American Hellenic Educational Progressive Association (AHEPA) University Hospital, Aristotle University of Thessaloniki, 54124 Thessaloniki, Greece

**Keywords:** Parkinson’s disease, inflammation, TNF-a, CCL2, MCP-1, biomarkers

## Abstract

One of the major mediators of neuroinflammation in PD is tumour necrosis factor alpha (TNF-α), which, similar to other cytokines, is produced by activated microglia and astrocytes. Although TNF-α can be neuroprotective in the brain, long-term neuroinflammation and TNF release can be harmful, having a neurotoxic role that leads to death of oligodendrocytes, astrocytes, and neurons and, therefore, is associated with neurodegeneration. Apart from cytokines, a wide family of molecules with homologous structures, namely chemokines, play a key role in neuro-inflammation by drawing cytotoxic T-lymphocytes and activating microglia. The objective of the current study was to examine the levels of the serum TNF-α and CCL2 (Chemokine (C-C motif) ligand 2), also known as MCP-1 (Monocyte Chemoattractant Protein-1), in PD patients compared with healthy controls. We also investigated the associations between the serum levels of these two inflammatory mediators and a number of clinical symptoms, in particular, disease severity and cognition. Such an assessment may point to their prognostic value and provide some treatment hints. PD patients with advanced stage on the Hoehn–Yahr scale showed an increase in TNF-α levels compared with PD patients with stages 1 and 2 (*p* = 0.01). Additionally, the UPDRS score was significantly associated with TNF-α levels. CCL2 levels, however, showed no significant associations.

## 1. Introduction

The second most prevalent progressive neurodegenerative disorder, Parkinson disease (PD), presents with a variety of clinical symptoms, including bradykinesia, tremor, cognitive impairment, and depression [1]. Although the exact cause of PD is yet to be discovered, it is known that the loss of the dopaminergic neurons in substantia nigra (SN) is the primary pathology underlying the gross motor symptoms of PD. Recent research has linked the pathogenesis of PD to a strong inflammatory response that is defined by brain microglia activation induced by higher cytokine levels [2,3].

In previous studies, it was shown that the cerebrospinal fluid (CSF) and nigrostriatal DA areas in PD patients had significantly higher amounts of cytokines, such TNF-α, interleukin-1 (IL-1), interleukin-2 (IL-2), interleukin-4 (IL-4), and interleukin-6 (IL-6) [4]. Additionally, glial cells expressing interferon-γ (INF-γ), TNF, and IL-1 were found in greater density in the SN of PD patients [5,6]. These cells, seen in the SN and striatum of PD patients but not in controls, are activated microglial cells producing proinflammatory cytokines which, in turn, increase the expression of specific enzymes, such as inducible NO synthase (iNOS) and cyclooxygenase-2 (COX-2) [3]. In addition to the nigrostriatal area, the hippocampus and cerebral cortex of PD patients were also characterized by activated microglia [7]. Microglia represent the resident macrophages of the brain. They are essential for modulating neurogenesis, they are involved in synaptic remodeling, and they regulate neuroinflammation by surveying the brain microenvironment [8]. Dysfunction of microglia has been observed and correlated to the onset and progression of several neurodevelopmental and neurodegenerative diseases [4]; however, the multitude of factors and signals that influence microglial activity have not been fully elucidated. This is because microglia not only respond to local signals within the brain, but also they receive input from the periphery, including the gastrointestinal (GI) tract [9]. Recent studies suggest that the gut microbiome plays a pivotal role in the regulation of microglia maturation and function. Moreover, altered composition of the intestinal microbial community has been reported in neurological disorders with known microglial involvement in humans [10,11].

One of the major mediators of neuroinflammation in PD is TNF-α, which, similar to other cytokines, is produced by activated microglia and astrocytes. Although TNF-α can be neuroprotective in the brain, long-term neuroinflammation and TNF-α release can be harmful, having a neurotoxic role that leads to death of oligodendrocytes, astrocytes, and neurons and, therefore, is associated with neurodegeneration [12,13]. Moreover, a sustained inflammation accompanied by systemic activation of the immune system is observed in PD patients, with changes in the periphery, specifically in cells of the innate immune system, e.g., monocytes, and the adaptive immune system, such as T lymphocytes [14].

Apart from cytokines, a wide family of molecules with homologous structures, namely chemokines, play a key role in neuro-inflammation by drawing cytotoxic T-lymphocytes and activating microglia [15]. PD, Alzheimer’s disease, and amyotrophic lateral sclerosis are three examples of neurodegenerative disorders that may develop as a result of altered levels of CCL2 expression in the microglia and/or neurons [16,17,18]. According to Reale et al., the pathophysiology of neurodegeneration may be connected to the elevated serum level of CCL2 in PD patients [19]. Nevertheless, there are controversial findings regarding serum CCL2 levels in PD patients.

The objective of the current study was to examine the levels of the serum TNF-α and CCL2 in PD patients and healthy controls. We also investigated the associations between the serum levels of these two inflammatory mediators and a number of clinical symptoms, in particular, disease severity and cognition. Such an assessment may point to their prognostic value and provide some treatment hints.

## 2. Results

### 2.1. Demographic and Clinical Characteristics

The PD group of 142 patients had an average age of 65 years, with an age range between 36 and 83 years; this was not significantly different from the control group of 67 volunteers (mean age 65 and range of 30 to 88 years). The mean age at onset of Parkinson’s disease was 62 years, with a range 36–82 years. There were also no significant sex differences between the groups (Fisher’s exact test, *p* = 0.43). More demographic and clinical data are described in Table 1.

### 2.2. Serum Inflammatory Factors in PD and Control Groups

PD patients had significantly higher serum TNF-α (*p* = 0.001) levels compared with the controls. The mean levels of serum TNF-α were 88.56 pg/mL (21.3–730.9, SD 78.17) in patients and 63.24 pg/mL (24–221 SD 33.33) in controls. The median level of CCL2 in the serum of PD patients did not significantly differ from that of the control group (*p* = 0.3). The mean levels of serum CCL2 were 190.76 pg/mL (39.94–190.76, SD 173.88) in patients and 207.75 pg/mL (72.38–994.25 SD 161.84) in controls (Figure 1). There was no significant association found between age, age at onset, and sex and the level of cytokines in the PD and control groups (r values were in the range from −0.07 to +0.15, *p* > 0.05). None of the analyzed cytokines reached significance with regard to motor subtyping. Several other comorbidities, such as diabetes, hypertension, dyslipidemia, and smoking, were examined for a possible association with serum TNF-α and CCL2 levels, without any significant results.

### 2.3. The Link between Inflammatory Factors and PD Severity

Correlations were found between PD severity and the levels of TNF in serum. In particular, PD patients with advanced stage on the Hoehn–Yahr scale showed an increase in TNF-a levels compared with PD patients with stages 1 and 2 on the Hoehn–Yahr scale (*p* = 0.01). Additionally, UPDRS score was also significantly associated with TNF levels, *p* < 0.0001 (Figure 2) CCL2 levels, however, showed no significant associations.

### 2.4. Cognitive Decline and Motor Phenotypes in PD and Inflammatory Factors

There were not any significant correlations between the serum levels of TNF-a and cognitive decline measured with MMSE and ACEIII (Figure 3). Although cognitive impairment in Parkinson’s disease patients was strongly associated with the global severity of PD measured by H and Y stage and motor symptoms of PD measured by UPDRS part III, linear multiple regression analysis showed that increase in TNF serum levels is not associated with severity of cognitive decline (*p* > 0.05). CCL2 levels also showed no significant associations.

### 2.5. Main Association of PD with Cytokine Levels from Previous Analysis

The main outcome of TNF-α and MCP-1 levels is presented in Table 2 for each study.

In total, 23 case control studies for TNF-α and only three studies for MCP-1 were found in the literature. The average number of patients who were examined in these studies was 68 patients (range 8–230). The majority of studies showed a positive correlation with disease in general, with disease severity, with cognitive decline, and with other neuropsychiatric disorders in the context of Parkinson’s disease, such as depression and anxiety. MCP-1 was less studied in the literature, with positive association in the majority of studies but with a very limited number of patients in each one (average number of patients 23, range 10–47)

## 3. Discussion

In recent years, a growing body of epidemiological and genetic research, as well as assessments of post-mortem and biofluid markers, have shown that inflammation can have an important impact on Parkinson’s disease incidence and development. Despite the obvious role that inflammation plays in the etiology of PD, there are still a number of unanswered questions that need to be resolved in order to prepare for clinical trials that try to modify the immune system in order to treat disease: What proof do we have that people with sporadic and genetic PD have specific pro-inflammatory patterns in their blood and CSF? Are there any differences in the blood/CSF inflammatory profiles during the course of the disease? Which inflammatory markers in the blood/CSF offer the best chances for patient stratification, cohort enrichment, and outcome measurements in clinical trials?

Peripheral blood sampling is a minimally invasive method and, therefore, measurement of immunological markers in blood has frequently been used to examine the differences between PD patients and healthy controls. Peripheral levels of several immune markers appear to be elevated in PD patients in several studies, indicating a continuing inflammatory process that may promote, or at least accompany, neurodegeneration, despite discrepancies in study design, analysis technique, and conclusions in these investigations [19].

Additionally, some of these investigations looked at relationships between marker levels and various clinical traits, such motor deterioration, disease severity, and non-motor status [28]. A further concern was posed regarding the possibility that a significant increase in already present inflammation may be related to non-motor characteristics such as autonomic dysfunction or dementia, even though the data are ambiguous [28,29].

In line with earlier research, we showed a strong association between serum levels of TNF-α and Parkinson’s disease and, more specifically, with the severity of the disorder. A meta-analysis that was undertaken in 2016 showed significant peripheral blood pro-inflammatory cytokine levels, including TNF-α in patients with PD compared with controls [30]. Since then, several other studies have confirmed these findings, and they highlighted a further association with disease severity, cognitive decline, and several other neuropsychiatric disorders in PD, such as depression or anxiety [21,28,31,32]. However, there were a few studies showing significant lower levels of TNF-α in patients compared with controls [22,23].

CCL2, also known as MCP-1, is a peripheral marker less studied compared with TNF-α. Our research did not reveal any significant associations with PD or other phenotypical trajectories in PD. Four studies have been contacted so far with opposing results. Increased MCP-1 serum levels compared with age-matched controls were described in two studies in PD patients [33,34]. A recent study reported no significant differences in serum but a significant increase of CCL2 in the CSF of PD patients [35]. Furthermore, reduced plasma levels of CCL2 were identified in patients with mutations in the *GBA* gene that encodes the enzyme beta-glucocerebrosidase, in comparison with sporadic Parkinson’s disease patients and controls [36].

There were a few restrictions in this study, such as the sample size that was relatively small, which has made it difficult to find any links between PD and substantial peripheral immunological markers. However, most studies reported to date had fewer participants compared with this one. Second, only two inflammatory markers were included in the study, and, therefore, the full picture of the immunological responses is missing. Additionally, this is not a longitudinal study; therefore, it is yet unclear how changes over time in peripheral inflammatory markers relate to the development of the illness and other NMSs. As the disease progresses, the pace of inflammation and, consequently, the level of indicators in the blood may change because the inflammatory response may alter in response to changes in the number of cells alive, the number of receptors, or the buildup of alpha-synuclein. It has been shown that the inflammatory response varies among various brain regions and that there are up- and downregulations in the mRNA expression of pro-inflammatory markers between early and late Braak stages [37]. Additionally, the activation of microglia by α-synuclein may upregulate both their toxic and neuroprotective activity, thus restricting the ability to interpret immunological signals in PD [38,39]. Depending on the degree of inflammation, immune markers may potentially operate in both pro- and anti-inflammatory manners. A growing body of research suggests that some cytokines, including well-known pro-inflammatory cytokines, such as TNF-, IFN-, or IL-6, may have a dual function [29,30,31,32], supporting the argument that classifying cytokines as pro- or anti-inflammatory is oversimplified and may not accurately depict the actual inflammatory landscape.

Serum immune marker levels are also influenced by a number of other factors, such as diet, body mass index, sleep, smoking, thyroid hormone levels, exercise, coffee use, frailty, or depression, and can, therefore, be regarded as non-specific [28,29]. These elements might have influenced how this study resulted [40,41,42,43,44]. Therefore, results from such studies should be evaluated with the proper caution in light of all these uncontrollable factors.

Another limitation is an underlying infection that cannot be excluded at the time of investigation. We may have been able to exclude patients with an active infection by excluding patients with a blood CRP level >1, but for a better way to identify people with an acute infection, serum levels of additional positive or negative acute-phase proteins, such as procalcitonin, alpha-1 antitrypsin, albumin, or ferritin, might have also been taken into consideration. Furthermore, the only potential confounders we considered were age, sex, disease duration, smoking, cardiovascular diseases, and diabetes. Other elements including body mass index, exercise, sleep, and other possible comorbidities were not considered.

Moreover, the substantial diversity of immunological markers may be influenced by intrinsic factors, including treatment and the development of the disease [45,46]. The impact of antiparkinsonian therapy on persistent inflammation is poorly understood. Dopamine may have some influence on the immunological response, according to a number of investigations [47]. On the basis of these results, some studies investigated the impact of therapy on immunological markers [48].

In conclusion, our findings indicate that the severity of PD is related to increased peripheral inflammation, which is primarily mediated by TNF-a. To more precisely characterize the specific involvement of peripheral inflammation in the development of motor and NMSs in PD, additional research on large samples with a wider variety of cytokines is necessary.

## 4. Materials and Methods

### 4.1. Study Participants

In this case–control study, 142 patients with PD and 67 healthy volunteers were enrolled at the outpatient clinic of General Hospital of Larisa, Thessaly, Greece. The diagnosis of PD was fulfilled in accordance with clinical diagnostic criteria of the UK Parkinson Disease Society Brain Bank. The exclusion criteria were any other chronic inflammatory/autoimmune diseases and patients with a serum CRP level >1. Among data collection were included demographic information such as sex, age, disease duration, age at onset, and family history. Healthy controls, with an overall age and sex distribution similar to that of the patients, were recruited from the patients’ family (spouses) or from individuals who were willing to participate in the research. The same exclusion criteria were applied to the controls. Our study was approved by the local Ethics Committee 43/08.07.2022, and informed consent was obtained from all participants.

Baseline assessments included medication history and comorbid conditions. The severity of the disease was assessed by modified Hoehn and Yahr staging (H and Y) [24]. Subjects were assessed with a number of standardized instruments, including the Movement Disorder Society–revised Unified Parkinson’s Disease Rating Scale (MDS-UPDRS) [25], Mini Mental State Examination(MMSE) [26], and Addenbrooke’s Cognitive Examination III, (ACE-III) [27]. PD patients were divided into Tremor Dominant (TD), Akinetic/Rigid (AR), and Mixed (MX) subtypes on the basis of the criteria used in previous studies [20].

### 4.2. Blood Sample and ELISA

Peripheral venous blood samples were collected from the patients and healthy controls. Blood samples were allowed to clot and then centrifuged at 3500 rpm for 10 min. The supernatant (serum) was separated and stored immediately at −80 °C. Serum levels of TNF and CCL2 were measured by commercially available enzyme-linked immune sorbent assay (ELISA) kits. Specifically, we used the human TNF-a ELISA Kit (Invitrogen, Waltham, MA, USA), which has analytical sensitivity of 1.7 pg/mL and whose detection range is 16.5–1000 pg/mL), and the human CCL2/MCP-1 Quantikine ELISA kit (R&D Systems, Minneapolis, MN, USA), with analytical sensitivity of 10 pg/mL and detection range 31.3–1000 pg/mL.

### 4.3. Statistical Analysis

The collected data were coded, tabulated, and statistically analyzed using SPSS statistics (Statistical Package for Social Sciences) software version 26.0. Simple descriptive statistics (arithmetic mean and standard deviation, SD) were used for summary of normal quantitative data, and frequencies were used for qualitative data. Bivariate relationships were displayed in cross-tabulations, and comparison of proportions was performed using the Chi-square and Fisher’s exact tests, where appropriate. The serum level of TNF-α and CCL2 were analyzed by independent t and chi-square tests. An independent samples test was used in order to examine the differences between serum levels of cytokines and chemokines in patients and controls. Correlations between variables were calculated by Pearson’s correlation coefficient, and simultaneous effects of various factors on these cytokines were analyzed by multiple linear regressions with backward method. *p* < 0.05 was considered statistically significant. Box plot and scatter plot graphs were used to depict the correlations.

### 4.4. Search Strategy

We conducted a systematic review of the literature (G.X, C.M.), investigating all studies about peripheral blood inflammatory markers that included TNF-α and CCL2 and Parkinson’s disease. We performed searches on PubMed, Cochrane Library, and MEDLINE (via PubMed) to identify all published studies before October 2022. A combination of the terms “inflammatory biomarkers”, “TNF-α,” “TNF,” “CCL2”, “MCP-1”, and “Parkinson’s disease” were used. All studies presenting original data that reported the clinical and inflammatory characteristics of patients with Parkinson’s disease were included for further review. English language was the only filter used initially. References of selected articles and reviews were also searched for additional records (Figure 4).

Data extracted from each study were title, author, year of publication, the exact number and type of inflammatory biomarkers that were investigated, and the association that was found.

### 4.5. Data Collection and Eligibility Criteria

Studies were selected when they provided information on peripheral cytokine concentrations in PD patients and healthy controls and when they met the following criteria: (1) case control studies, (2) languages being limited to English, and (3) full study. All in vitro studies that reported stimulated or unstimulated levels of cytokines were excluded from the analysis.

## Figures and Tables

**Figure 1 ijms-24-00264-f001:**
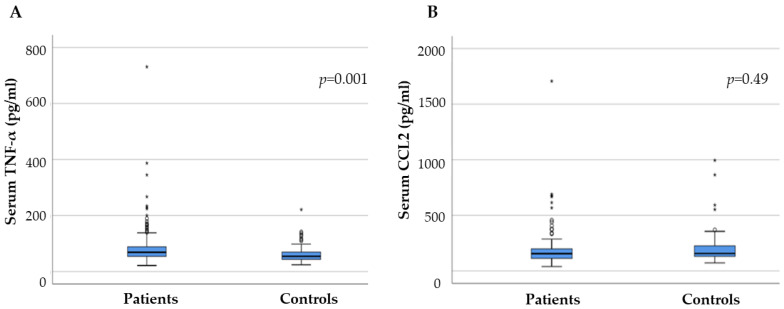
Serum levels of (**A**) TNF-a and (**B**) CCL2 in patients and controls. PD patients had significantly higher serum TNF (*p* = 0.001) levels compared with controls. The median level of CCL2 in the serum of PD patients did not differ significantly from the control group (*p* = 0.3). * represents the extreme values.

**Figure 2 ijms-24-00264-f002:**
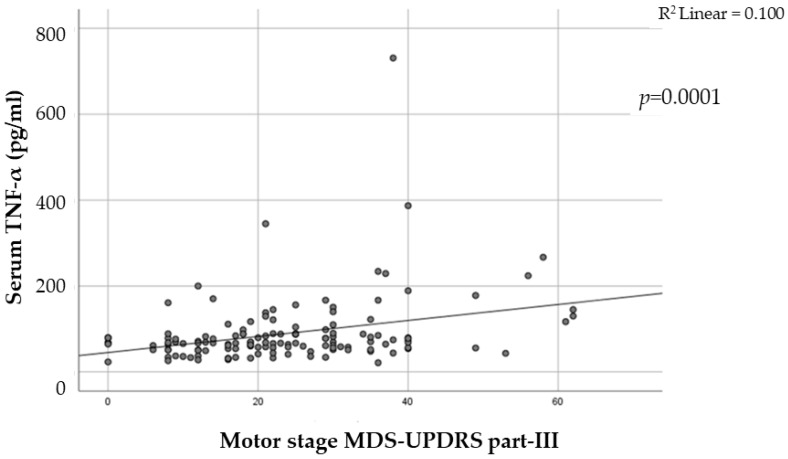
Linear multiple regression analysis showing the association between serum levels of TNF-a and disease severity measured by MDS-UPDRS part III. The higher the score in UPDRS part III, the higher the levels of serum TNF-a (*p* ≤ 0.0001).

**Figure 3 ijms-24-00264-f003:**
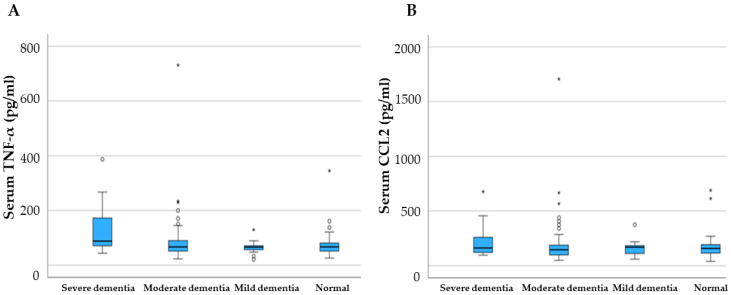
Cognitive decline measured with ACEIII in relation with serum levels of (**A**) TNF-α and (**B**) CCL2. TNF-α and CCL2 serum levels are not associated with severity of cognitive decline (*p* > 0.05). * represents the extreme values.

**Figure 4 ijms-24-00264-f004:**
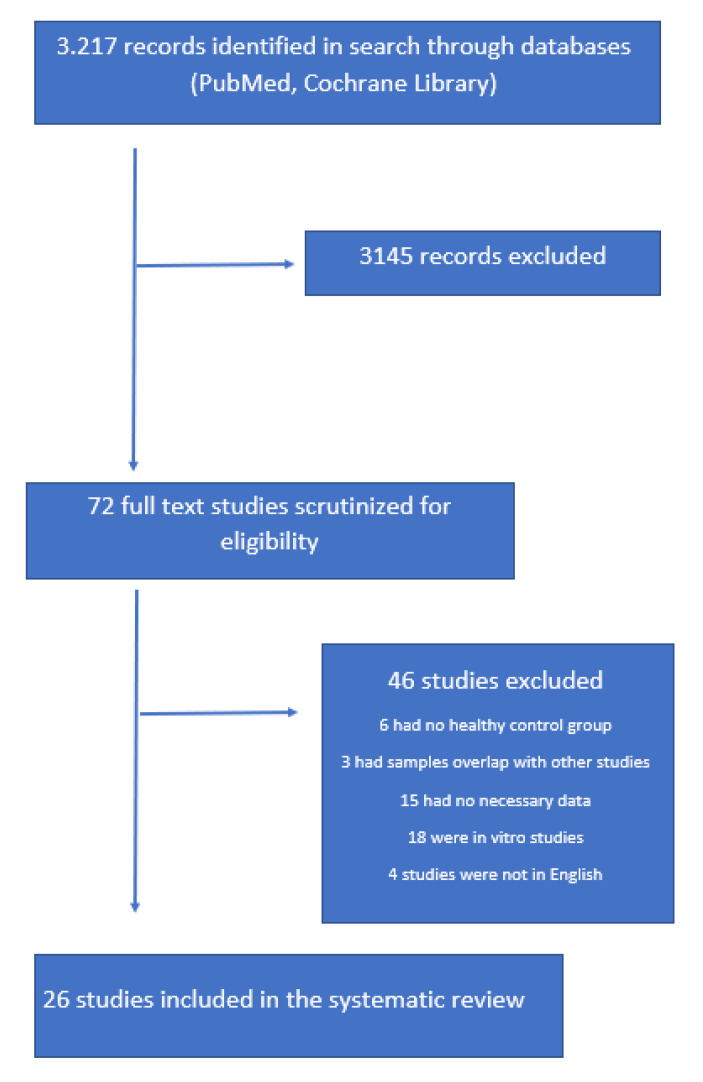
Prisma flowchart of the literature search according to eligibility and exclusion criteria that were set.

**Table 1 ijms-24-00264-t001:** Demographic and clinical data of PD patients enrolled in the study.

Demographics	PD Patients (142)
Mean age (range)	65 (36–83)
Disease duration (years) mean (SD)	7.5 (4)
Sex	M. = 87, F. = 55
PD phenotype	Tremor dominant 49%, PIGD 28%, intermediate 23%
H and Y Stage *	Mild 49%, moderate 34%, severe 14%
Motor stage MDS-UPDRS part-III **	Mild 49%, moderate 37%, severe 14%
Cognitive impairment MMSE ***	Normal 64%, mild 12%, moderate 20%, severe 4%
Family history	No 83%
Smoking	19%
Other comorbidities: Diabetes	16%
Hypertension	47%
Dyslipidemia	20%

* Stages of the disease were evaluated according to Hoehn–Yahr scale [20]. ** Motor examination was performed according to MDS-UPDRS [21]. *** For the cognitive evaluation of the patients, MMSE [22] and ACER-Addenbrooke [23] tests were used. The results according to MMSE are presented here.

**Table 2 ijms-24-00264-t002:** Summary of outcomes of measurement of peripheral TNF-a and MCP-1 levels in Parkinson’s disease.

Main Outcome	Studies *	PD Cases/Controls	*p*-Values
Lower TNF-α in cases versus controls	Choi et al., 2012 [1]	8/13	0.6
	Gupta et al., 2016 [2]	81/83	<0.001
	Eidson et al., 2017 [3]	12/6	<0.001
	Li et al., 2018 [4]	43/24	0.7
	Rocha et al., 2018 [5]	23/21	<0.001
Higher TNF-α in cases versus controls	Brodacki et al., 2008 [6]	48/20	<0.005
	Gruden et al., 2012 [7]	32/26	<0.005
	Koziorowski et al., 2012 [8]	60/24	<0.001
	Bu et al., 2015 [9]	131/141	0.002
	Hu et al., 2015 [10]	152/31	0.034
	Csencsits-Smith et al., 2016 [11]	15/24	<0.005
	Yang et al., 2018 [12]	120/100	<0.05
	Kim et al., 2018 [13]	58/20	>0.1
	King et al., 2019 [14]	156/64	<0.05
	Alrafiah et al., 2019 [15]	26/24	>0.01
	Li et al., 2022 [16]	138/132	<0.005
Correlation of TNF-α with disease severity (H and Y, UPDRSIII)	Dobbs et al., 1999 [17]	78/140	0.015
	Williams-Gray et al., 2016 [18]	230/93	<0.005
	Kouchaki et al., 2018 [19]	83/83	<0.0001
	Rathnayake et al., 2019 [24]	72/30	<0.001
Correlation of TNF-α with cognitive decline	Williams-Gray et al., 2016 [18]	230/93	<0.007
	Karpenko et al., 2018 [25]	117/60	<0.05
Correlation of TNF-α with depression and anxiety	Lindqvist et al., 2012 [26]	86/40	<0.001
	Wang et al., 2016 [27]	62/62	<0.005
Higher MCP-1 levels in cases versus control	Usenko et al., 2016 [20]	47/19	<0.001
	Csencsits et al., 2016 [11]	25/15	≤0.05
	Schröder et al., 2018 [28]	10/13	<0.05
Lower MCP-1 levels in cases versus control	Miluikhina et al., 2020 [29]	28/28	≤0.01

* All the included studies are listed in the supplementary data.

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
