# Peer review of "Peripheral Inflammatory Markers TNF-α and CCL2 Revisited: Association with Parkinson’s Disease Severity"

_ijms, 2022, doi:10.3390/ijms24010264_

Round 1

Reviewer 1 Report

In the current manuscript, the authors performed a perspective collection and clinical assessment of PD patients and healthy controls, combined with an assessment of the literature for studies reporting levels of TNF and CCL2 in patients with PD. The study is highly relevant, and adds to the existing literature examining the association of specific inflammatory markers and disease status. There are a few minor concerns that should be addressed before publication is recommended. 

1. In the Introduction, line 36, the authors should clarify this line, stating that the "dopaminergic neurons ... are the primary culprit"; it's perhaps better to state that the loss of these neurons are the primary pathology underlying the gross motor symptoms of PD, or something to that effect.

2. It is not clear to what measure the statistical p value in Figure 3 is referring to; and related to that, if it is indicating the strength of a correlation between TNF and UPDRS, this may be influenced strongly by the single outlier at the top of the plot.

3. Figure 4 should be re-labeled, as it is not displaying statistical correlations between TNF levels and cignitive decline.

4. In general, it would be helpful to have more detail provided in the figure legend for each Figure; particularly with respect to statistical tests shown. 

Author Response

We thank the reviewer for all  the comments and suggestions. 

1.We rephrased the sentence acccordingly.

2.The p value in Figure 3 depicts the association between serum levels of TNF-a and disease severity measured by MDS-UPDRS part III. The higher the score in UPDRS part III the higher the levels of serum TNF-a (p =<0.0001).We changed the legend in order to clarify the graph and the association

Even if we exclude from the analysis this outlier the p value remains significant p=0.004, although it seems to affect the statistical significance.The TNF levels of this extreme outlier is within the range that has been defined in other studies. 

3.We relabeled Figure 3 accordingly "Cognitive decline measured with ACEIII in relation with serum levels of TNF-a. TNF serum levels are not associated with severity of cognitive decline (p >0.05)"

4.We provided more detail and information in the figure legends 

Reviewer 2 Report

In this study, Xiromerisiou et al. investigated the correlation of the TNF and CCL2 levels in the serum of PD patients compared to healthy controls taking into account clinical symptoms. To measure TNF and CCL2 they used ELISA, a well-established method for analyte quantification in biological samples. The authors compared their data to similar studies and discussed limitation of their work. The methods section provides an adequate description and the results are presented well.

I only have a few minor comments.

1.       Please indicate the units in Figure 2,3,4.

2.       Introduce both names for CCL2 (CCL2 aka MCP-1) in the abstract so readers are not confused reading the main text.

3.       Proofread the text for typos, comma use, and grammar mistakes, I found quite some.

4.       In Methods, justify the use of the selected statistical methods.

5.       Results section P. 5 L. 153-157 – provide an exact value for fold difference in the level of the studied analytes.

6.       Table 1 – age range is missing.

Author Response

We thank the reviewer for the comments and suggestions

1.We added the units in the relevant figures, pg/ml

2.Thank you for this comment. We introduced both names for CCL2 (CCL2 and MCP-1) in the abstract so readers are not confused reading the main text.

3. We proofread the text for typos, comma use, and grammar mistakes

4.In methods, we described in a better way and we justified the use of the selected statistical methods

5.We added the following "PD patients had significantly higher serum TNF (p=0.001) levels compared to the controls. The mean levels of serum TNF -α in patients were 88,56 pg/ml (21,3-730,9, std dev 78,17) and in controls 63,24 pg/ml (24-221 std.dev. 33,33)"

6.We added the age range in table 1 for patients (36-89) st dev 11,19 

Reviewer 3 Report

The manuscript by Xiromerisiou G. et al tiltled as “Peripheral inflammatory markers, TNF and CCL2 revisited: association with Parkinson’s disease severity” highlights the association of serum level TNF-a and CCL2 (Chemokine [C-C motif] ligand 2) with PD disease clinical symptoms. The authors noted that serum TNF-a level increases in patients with the disease severity while the CCL2 level does not change significantly. These findings could be useful in the PD therapeutic strategy and diagnosis. The study is well designed, and data have been interpreted appropriately. However, the authors should explain why they investigated these two markers only and not the others. The article is well written overall.  

Suggested corrections:

1.    The figures/graphs quality can be improved. 

2.    TNF-a, CCL2 levels (at least average values) and their units should be given in the results and graphs.

3.    Figure.2 graph can be represented as ‘A and B’. 

4.    In M&M section, add a bit more on the TNF-a/CCL2 quantification method/protocol. 

5.    In Figure 3, CCL2 level is not shown (stated in line 170). 

6.    In Figure.4, an elaborative legend would be good. what does the asterisk (*) used indicate? Also, CCL-2 level graph is missing. 

7.    I could not find the Ref# 40 in main text, please check. 

Author Response

We thank the reviewer for the comments and suggestions

1.We tried to improve the quality of the figures as suggested

2.We thank the reviewer for this suggestion.We added the average values their units in the relults part and figures

3.  We presented figure 3 in A and B parts

4.  In M&M section,we  added more information on the TNF-a/CCL2 quantification method/protocol

5.We added figure 3 in the right context

6.The asterix indicates the extreme values.We added CCL2 graph and an elaborative legend

7. we added the reference 40 in the main text.